# Postoperative Pain after Different Transscleral Laser Cyclophotocoagulation Procedures

**DOI:** 10.3390/ijerph20032666

**Published:** 2023-02-02

**Authors:** Thomas Falb, Astrid Heidinger, Fabian Wallisch, Hrvoje Tomasic, Domagoj Ivastinovic, Marlene Lindner, Franz Tiefenthaller, Lukas Keintzel, Lukas Hoeflechner, Regina Riedl, Anton Hommer, Ewald Lindner

**Affiliations:** 1Department of Ophthalmology, Medical University Graz, 8036 Graz, Austria; 2Department of Dentistry and Oral Health, Medical University Graz, 8036 Graz, Austria; 3Institute for Medical Informatics, Statistics and Documentation, Medical University Graz, 8036 Graz, Austria; 4Department of Ophthalmology, Sanatorium Hera, 1090 Vienna, Austria

**Keywords:** glaucoma, transscleral cyclophotocoagulation, postoperative pain

## Abstract

Background: As the number of surgical options in glaucoma treatment is continuously rising, evidence regarding distinctive features of these surgeries is becoming more and more important for clinicians to choose the right surgical treatment for each individual patient. Methods: For this retrospective data analysis, we included glaucoma patients treated with either continuous wave (CW-TSCPC) or micropulse transscleral cyclophotocoagulation (MP-TSCPC) in an inpatient setting. Pain intensity was assessed using a numeric rating scale (NRS) ranging from 0 (no pain) to 10 (worst imaginable pain) during hospitalization. CW-TSCPC was performed using OcuLight^®^ Six (IRIDEX Corporation, Mountain View, CA, USA) and MP-TSCPC was performed using the IRIDEX^®^ Cyclo-G6 System (IRIDEX Corporation, Mountain View, CA, USA). Results: A total of 243 consecutive cases of TSCPC were included. Of these, 144 (59.26%) were treated with CW-TSCPC and 99 (40.74%) with MP-TSCPC. Using the univariable model, the risk for postoperative pain was observed to be lower in MP-TSCPC compared with CW-TSCPC (unadjusted: OR 0.46, 95% CI 0.24–0.84, *p* = 0.017), but this did not hold using the multivariable model (adjusted: OR 0.52, 95% CI 0.27–1.02, *p* = 0.056). Simultaneously conducted anterior retinal cryotherapy was associated with a higher risk for postoperative pain (OR 4.41, 95% CI 2.01–9.69, *p* < 0.001). Conclusions: We found that the occurrence of postoperative pain was not different in CW-TSCPC compared with MP-TSCPC in a multivariable model. In cases of simultaneous anterior retinal cryotherapy, the risk for postoperative pain was significantly higher.

## 1. Introduction

Although various risk factors for glaucoma development have been identified [1,2,3], intraocular pressure remains the only modifiable one [4]. While our current armamentarium for detecting glaucoma and glaucomatous progression has known limitations [5], emerging technologies may facilitate earlier diagnosis [6]. On the one hand, this is crucial for an effective glaucoma treatment; on the other, early treatment in eyes with mild glaucoma highlights the necessity to evaluate options carefully to maintain the old principle “primum non nocere”.

With the development of new surgical options, evaluating distinct features of available treatment options is of high interest for clinicians to personalize surgical glaucoma treatment for each individual patient. These features are not only limited to surgical success in terms of intraocular pressure control, but also involve complication rates and surgery-related patient discomfort. Knowledge about these outcome parameters is scarce, but can profoundly support clinicians in decision-making. In the end, patient-related life quality should be the ultimate aim of glaucoma management [7,8].

Intraocular pressure can be lowered using medication, laser or surgery. A reduction in intraocular pressure can be achieved by increasing the outflow of aqueous humor or by decreasing its production. Cyclodestructive procedures target the ciliary body, where aqueous humor is produced. Various methods have been used for cyclodestruction, including diathermy, cryotherapy, ultrasound and laser. Laser cyclophotocoagulation was first introduced in 1972 by Beckman and colleagues [9]. Different paths have been explored to approach the ciliary body, including transpupillary, transvitreal, endoscopic and transscleral. Transscleral cyclophotocoagulation (TSCPC) is a non-invasive treatment for glaucoma, in which the ciliary body is treated with laser energy through the intact sclera. There are various methods of TSCPC. In continuous-wave transscleral cyclophotocoagulation (CW-TSCPC), laser energy is applied as a continuous beam over a set period of time. It provides good efficacy and safety in different types of glaucoma, although severe potential side effects such as ocular hypotony, phthisis and vision loss may occur [10,11,12]. Micropulse transscleral cyclophotocoagulation (MP-TSCPC) uses an on/off method of laser application. Laser energy is applied as repetitive pulses spaced by time intervals without laser application. These intervals of non-application serve as cooling time and are supposed to reduce collateral damage to adjacent structures through overheating, thus reducing potential side effects [13,14]. Previous studies found a difference in safety favoring MP-TSCPC, which provides intraocular pressure reduction comparable with CW-TSCPC [15,16,17,18]. Traditionally, TSCPC was used for patients with neovascularization glaucoma or for patients with limited vision in whom previous filtration surgeries had failed. More recently, the use of TSCPC was extended to patients with earlier forms of glaucoma and good vision. The advent of MP-TSCPC has further contributed to this development, as its mode of action significantly limits tissue damage to the ciliary body. As the new method is supposed to offer improved tissue treatment, it should also reduce the frequency of side effects such as postoperative pain.

Postoperative pain is a common early postoperative complication after CW-TSCPC. Until now, pain after MP-TSCPC has been studied only as a secondary outcome parameter, with a tendency towards low or no pain postoperatively [18]. In this retrospective study, we aimed to compare early postoperative pain levels after CW-TSCPC and MP-TSCPC to clarify potential differences between these two procedures regarding the necessity of postoperative pain management. 

## 2. Materials and Methods

### 2.1. Patients

We retrospectively included all cases of CW-TSCPC and MP-TSCPC performed between January 2018 and December 2020 at the Medical University Graz. Exclusion criteria were insufficient or inconclusive data documentation as well as bilateral simultaneous treatment. 

### 2.2. Analysis

Electronic and paper records were retrieved for data analysis. Pain was assessed via a numerical rating scale (NRS) [19], a rating system with a scale from 0 (no pain) to 10 (worst imaginable pain). Assessment charts were filed for every patient upon inpatient admission until November 2018. Assessment charts were filed only if patients mentioned pain greater or equal to NRS 3 at any time during their hospital stay. Pain was assessed at regular intervals for all inpatients at the Medical University Graz. Other data collected included sex, age, type of glaucoma, laterality, preoperative intraocular pressure, total applied laser energy levels, and whether it was the first or a repetitive laser treatment. 

### 2.3. Surgical Technique

CW-TSCPC was performed using OcuLight^®^ Six (IRIDEX Corporation, Mountain View, CA, USA). In CW-TSCPC, laser energy is applied continuously over a fixed time. The desired effect is cyclodestruction with subsequent reduction of aqueous production. Settings were either fixed through the procedure, or adjusted until audible feedback (“plop”) was recognized. The approach used depends on a surgeon’s preferences. MP-TSCPC was performed using the IRIDEX Cyclo-G6 System (IRIDEX Corporation, Mountain View, CA, USA). In MP-TSCPC, laser energy is applied in an on/off fashion, with pulses of laser energy interrupted by pauses, which serve as cooling time and aim to reduce collateral damage. All procedures were performed under retrobulbar anesthesia in an operating theatre and inpatient setting. Lidocaine 2% was used for retrobulbar anesthesia. Patients experiencing pain during the operation received Alfentanil, which is a synthetic opioid analgesic drug that acts fast and lasts for approximately 15 min. Patients stayed in hospital overnight and were released on the first postoperative day. During this period, patients were monitored for post-operative pain. Until September 2019, the majority of operations were CW-TSCPC, thereafter MP-TSCPC. Total applied laser energy was calculated in Joule for MP-TSCPC (total applied energy (J) = watt × treatment duration in seconds × duty cycle) and CW-TSCPC (total applied energy (J) = watt × treatment duration per spot in seconds × total count of treated spots). Anterior retinal cryotherapy was performed using Keeler Cryomatic MKII (Keeler, Windsor, UK). Twelve spots at −80 °C for 15 s were applied, three spots in each quadrant of the treated eye. Anterior retinal cryotherapy is performed in cases of proliferative retinopathy, where panretinal photocoagulation is impossible due to media opacities [20,21].

### 2.4. Statistics

In descriptive statistics, continuous parameters are presented as mean and standard deviation (SD) or median, minimum and maximum. Categorical parameters are presented as frequencies and percentages. To evaluate the occurrence of postoperative pain equal to or greater than NRS 3 between the MP-TSCPC and CW-TSCPC, generalized estimating equations (GEE) with logit as link function and working independence correlation structure were used, accounting for repeated surgeries on the same patient [12]. The models were adjusted for patients’ sex, age, preoperative intraocular pressure, simultaneous conducted anterior retinal cryotherapy, laterality, previously performed TSCPC and total applied laser energy. Results are presented as odds ratios (ORs) with their corresponding 95% confidence intervals (CIs) and *p*-values. In sensitivity analyses, the models were repeated, including only the first documented surgery per patient. A *p*-value of *p* < 0.05 indicates statistical significance. Statistical analyses were performed using SAS 9.4 (SAS Institute, Cary, NC, USA).

## 3. Results

### 3.1. Patient Demographics and Clinical Characteristics

In total, 259 cases of TSCPC were identified. After excluding 16 patients with insufficient or inconclusive documentation or simultaneous bilateral treatment, 243 cases of TSCPC in 200 patients remained. Of these, 144 were treated with CW-TSCPC (63 female (43.8%); mean age, 71.4 years; SD, 15.6) and 99 were treated with MP-TSCPC (52 (52.5%) female; mean age, 68.8 years; SD, 16.0) (Table 1). 

Glaucoma subtypes treated were: neovascular glaucoma (59 (41.0%) in the CW-TSCPC group and 37 (37.4%) in the MP-TSCPC group), pseudoexfoliation glaucoma (25 (17.4%) in the CW-TSCPC group and 22 (22.2%) in the MP-TSCPC group), primary open angle glaucoma (18 (12.5%) in the CW-TSCPC group and 15 (15.2%) in the MP-TSCPC group), secondary glaucoma due to surgery, inflammation or trauma (27 (18.8%) in the CW-TSCPC group and 18 (18.2%) in the MP-TSCPC group), and other, less frequent glaucoma subtypes, including but not limited to pigmentary glaucoma, aphakic glaucoma and glaucoma due to various syndromes. Simultaneous anterior retinal cryotherapy was performed in 49 cases (35 (24.3%) in the CW-TSCPC group and 14 (14.1%) in the MP-TSCPC group) and simultaneous intravitreal injection was performed in 5 cases (4 (2.8%) in the CW-TSCPC group and 1 (1.0%) in the MP-TSCPC group). Preoperative pain greater than or equal to NRS 3 in the affected eye was mentioned in 5 cases (2 (1.4%) in the CW-TSCPC group and 3 (3.0%) in the MP-TSCPC group). 

### 3.2. Postoperative Pain

Postoperative pain greater than or equal to NRS 3 was reported in 52 cases (38 (26.4%) after CW-TSCPC and 14 (14.1%) after MP-TSCPC). Pain intensity, assessed via maximum NRS values in cases with relevant postoperative pain (NRS ≥ 3) was comparable between groups (4.8 (SD 1.1) in the CW-TSCPC group and 5.1 (SD 1.4) in the MP-TSCPC group). 

Using the univariable model, the risk for postoperative pain was observed to be lower for MP-TSCPC compared with CW-TSCPC (unadjusted: OR 0.46, 95% CI 0.24–0.84, *p* = 0.017), but this did not hold using the multivariable model (adjusted: OR 0.52, 95% CI 0.27–1.02, *p* = 0.056) (Table 2). Simultaneous conducted anterior retinal cryotherapy was associated with a higher postoperative pain risk (OR 4.41, 95% CI 2.01–9.69, *p* < 0.001). Of 52 patients with postoperative pain, 22 were treated using simultaneous anterior retinal cryotherapy. When cases with combined anterior retinal cryotherapy were excluded, the risk for postoperative pain was not different between the laser groups (OR 0.67, 95% CI 0.30–1.49, *p* = 0.331). Older age was protective (OR 0.98, 95% CI 0.96–1.00, *p* = 0.047). Sex, laterality, previously performed TSCPC, total applied laser energy and preoperative intraocular pressure did not show statistically significant impacts upon occurrence of pain equal to or higher than NRS 3. No difference in intraocular pressure on the first postoperative day was found between patients experiencing pain with NRS ≥ 3 and other patients (23.8 [9.1] versus 22.3 [9.8] mmHg, respectively; *p* = 0.310). 

## 4. Discussion

In this retrospective data analysis, we evaluated postoperative pain after CW-TSCPC and MP-TSCPC. Postoperative pain greater than or equal to NRS 3 was reported after 26.4% of CW-TSCPCs and after 14.1% of MP-TSCPCs. Nevertheless, using a multivariable model we did not find a laser variant to be a significant risk factor for postoperative pain. Combined anterior retinal cryotherapy and age were significant contributors to the occurrence of postoperative pain. 

In our study, the largest group of patients had neovascular glaucoma. Other studies investigating transscleral cyclophotocoagulation had varied study population compositions in terms of the type of glaucoma and favoring, e.g., primary open angle glaucoma [22]. This may be due to different types of lasers. With the advent of MP-TSCPC, there has been a trend toward using TSCPC in earlier stages of glaucoma, as studies have shown that MP-TSCPC is associated with substantially less severe complications, including hypotony [23].

Quigley at al. [10] investigated postoperative pain after CW-TSCPC as a secondary parameter in a retrospective chart review of 70 eyes using a scale from 0 to 10. Pain from 1–3 was reported in 29% of cases, 4–7 in 7% of cases and 8–9 in 9% of cases after CW-TSCPC performed using retrobulbar anesthesia. Aquino et al. [15] investigated postoperative pain after CW-TSCPC versus MP-TSCPC in 43 patients (22 after MP-TSCPC and 21 after CW-TSCPC). Postoperative pain was evaluated 1 week after surgery and was reported exclusively after CW-TSCPC; however, a statistically significant difference was not found between groups. In the study by Aquino et al., 62.6 Joules were delivered in total via MP-TSCPC and 60–112 Joules were applied via CW-TSCPC. In our study, the total applied laser energy was 92.0 Joules via CW-TSCPC and 97.9 Joules via MP-TSCPC. Despite a higher applied laser energy, there was a trend of less postoperative pain after MP-TSCPC, which might have become significant if equal energies had been used for both laser types. Fili et al. [16] investigated postoperative pain as a secondary outcome parameter in 30 eyes (15 per group) of 22 patients after MP-TSCPC and CW-TSCPC performed under general anesthesia. They noted that no postoperative ocular pain was reported in the MP-TSCPC group, but 12 patients in the CW-TSCPC group indicated low-grade pain. In our study, we did not document NRS scores below three in a large percentage of patients, so we may have missed a difference in this regard. In their prospective study about postoperative pain after different glaucoma surgeries, Li et al. [17] included 15 cases of CW-TSCPC. Postoperative pain was assessed via NRS scale at multiple time points within 24 h after surgery and regarded as clinically significant if at least NRS 5 was reported; of 15 patients, 14 (93.3%) experienced pain after cyclophotocoagulation. The odds ratio for experiencing significant postoperative pain after cyclophotocoagulation in comparison to trabeculectomy was 30.9 (95% CI 3.47–275.1; *p* = 0.002). Tan et al. [18] reported mild postoperative pain without necessity of pain medication in 7 (18.4%) of their 38 patients after MP-TSCPC treatment on the first postoperative day. Another study by Chang et al. reported mild postoperative pain in 5.8% of cases [24]. This was considerably lower than in our study, which might be due to close surveillance of our patients in an inpatient setting. 

Here, we found that older patients experienced less postoperative pain. The mean age of our patients was approximately 70, which was considerably higher than in other studies [10,15,16]. In a large retrospective study including more than 10,000 patients from 26 countries, it was shown that postoperative pain decreases with increasing age [25]. Many elderly patients believe that pain is a normal part of life and thus are more resilient, which means that they adapt positively when faced with the adversity of pain [26].

The highest postoperative pain in our study was associated with simultaneously conducted anterior retinal cryotherapy. Studies about postoperative pain after anterior retinal cryotherapy are scarce. Steel et al. [27] reported pain after retinal cryopexy for retinal breaks in only 3 [SD ± 1.6] out of 60 patients. However, anterior retinal cryotherapy of four quadrants in neovascular glaucoma has a relatively larger area of treatment compared to focal retinal cryopexy for retinal breaks. 

This study has several limitations, with its retrospective nature being one of the most important. Pain is influenced strongly by cultural factors and should be investigated in different populations. Reporting pain can be influenced by mental diseases, which were not included in our analysis. We did not investigate other comorbidities causing pain. As the mean age was approximately 70 years, several diseases, e.g., degenerative disorders of the spine, may have contributed to the intensity of pain. Still, as age was not different, this should have affected both groups in the same way. We did investigate preoperative pain, and interestingly this was quite rare in both groups, with 1.4% and 3% in the CW-TSCPC group and the MP-TSCPC group, respectively. Finally, we did not assess treatment success. Whether a sufficient laser energy application in terms of treatment success would also lead to a higher rate of postoperative pain remains unclear. 

Glaucoma and glaucoma therapy put a financial strain on healthcare systems worldwide [28,29,30,31,32,33]. Hospitalization is one of the major cost drivers in glaucoma treatment. TSCPC is performed in an inpatient or outpatient setting, depending on local hospital policies. With our study, we hope to provide evidence regarding postoperative pain after transscleral cyclophotocoagulation as a potential support for decision making regarding hospitalization and pain medication for TSCPC. 

## 5. Conclusions

Our data do not indicate a difference in postoperative pain between MP-TSCPC and CW-TSCPC. Postoperative pain was shown to be highest for simultaneously conducted anterior retinal cryotherapy. Patients should be informed about postoperative pain as part of the informed consent, especially in cases with combined anterior retinal cryotherapy. Postoperative pain can be managed with on-demand pain medication.

## Figures and Tables

**Table 1 ijerph-20-02666-t001:** Patient demographics and clinical characteristics. Values are given as n (%) unless otherwise indicated.

	CW-TSCPC	MP-TSCPC	*p*-Value
Age (years, mean (SD))	71.4 (15.6)	68.8 (16.0)	0.235
Sex (male)	81 (56.3%)	47 (47.5%)	0.227
NVG	59 (41.0%)	37 (37.4%)	0.773
POAG	18 (12.5%)	15 (15.2%)	
PEXG	25 (17.4%)	22 (22.2%)	
Secondary Glaucoma	27 (18.8%)	18 (18.2%)	
Others	15 (10.4%)	7 (7.1%)	
Previous TSCPC	25 (17.4%)	27 (27.3%)	0.082
Preoperative IOP (mmHg, mean (SD))	35.2 (12.2)	34.3 (10.4)	0.574
Right Eye	68 (47.2%)	35 (35.4%)	0.102
Anterior Retinal Cryotherapy	35 (24.3%)	14 (14.1%)	0.061
Total Applied Energy (Joule, mean (SD))	92.0 (33.7)	97.9 (12.3)	0.114

CW-TSCPC—continuous wave transscleral photocoagulation; MP-TSCPC—micropulse transscleral cyclophotocoagulation; SD—standard deviation; NVG—neovascular glaucoma; POAG—primary open angle glaucoma; PEXG—pseudoexfoliation glaucoma; IOP—intraocular pressure.

**Table 2 ijerph-20-02666-t002:** Occurrence of postoperative pain equal to or higher than NRS 3 for MP-TSCPC and CW-TSCPC.

Postoperative Pain NRS ≥ 3	Odds Ratio	95% Lower CI	95% Upper CI	*p*-Value
Univariabel model				
Laser variant	0.46	0.24	0.84	0.017
Multivariable model				
Laser variant	0.52	0.27	1.02	0.056
Age	0.98	0.96	1.00	0.047
Sex	1.10	0.57	2.14	0.774
Previous TSCPC	0.76	0.27	2.10	0.596
Preoperative IOP	1.03	1.00	1.06	0.055
Laterality	1.07	0.54	2.11	0.841
Anterior retinal cryotherapy	4.41	2.01	9.69	<0.001
Total applied laser energy	1.00	0.99	1.01	0.626

CW-TSCPC—continuous wave transscleral photocoagulation; MP-TSCPC—micropulse transscleral cyclophotocoagulation; NRS—numerical rating scale; IOP—intraocular pressure.

## Data Availability

The datasets generated and/or analyzed during this study are available from the corresponding author upon reasonable request.

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
