# Peer review of "Postoperative Pain after Different Transscleral Laser Cyclophotocoagulation Procedures"

_ijerph, 2023, doi:10.3390/ijerph20032666_

Round 1

Reviewer 1 Report

The authors present a retrospective analysis on 243 patients undergoing two different laser procedures to treat glaucoma. They developed a multivariable model to assess if one of the two techniques was associated to a higher risk of developing post-operative pain. 

They found that the risk for postoperative pain was not different in the two groups of patients.

The study presents several limitations, well underlined by the authors (retrospective design, no assessment of treatment success).

Here some minor comments: 

Acronyms included in table 1 and 2 should be explained. 

I have some questions: 

-       Which local anesthetics was used for retrobulbar anesthesia?

-       It is not clear for me during which period the patients were monitored for detecting post-operative period.

-       Did the authors administer analgesic drugs during the procedures? Did they use a rescue therapy for patients experiencing pain?

-       The authors declare that patients with a well-controlled pain were excluded for the absence of data; how many are these patients? 

Thank you

Author Response

We thank the reviewers for their appreciating comments and for their suggestion, which helped to improve our work. We did our best to address all points appropriately and we hope that you find our manuscript now suitable for publication.

Acronyms included in table 1 and 2 should be explained. 

Answer: Acronyms in table 1 and 2 were explained.

Which local anesthetics was used for retrobulbar anesthesia?

Changes: (Methods) Lidocaine 2% was used for retrobulbar anesthesia.

It is not clear for me during which period the patients were monitored for detecting post-operative period.

Changes: (Methods) Patients stayed overnight and were released on the first postoperative day. During this period patients were monitored for post-operative pain.

Did the authors administer analgesic drugs during the procedures? Did they use a rescue therapy for patients experiencing pain?

Changes: (Materials and Methods) Patients experiencing pain during the operation received Alfentanil, which is a synthetic opioid analgesic drug that acts fast and lasts for about 15 minutes.

The authors declare that patients with a well-controlled pain were excluded for the absence of data; how many are these patients? 

Answer: We did not exclude patients with a well-controlled pain. However, we excluded patients with insufficient or inconclusive data documentation as well as bilateral simultaneous treatment. In total, 16 patients were excluded.

Changes: (Results) After excluding 16 patients with insufficient or inconclusive documentation or simultaneous bilateral treatment, 243 cases of TSCPC in 200 patients remained.

Reviewer 2 Report

Postoperative Pain After Different Transscleral Laser 2 Cyclophotocoagulation Procedures

This a retrospective study that compares post-operative pain intensity in glaucoma patients treated with either continuous wave (CW-TSCPC) or micropulse transscleral cyclophotocoagulation (MP-TSCPC) in an inpatient setting. Using a multivariate model, the study reports that post-operative pain was not significantly different in both cohort of patients.

General concerns: This study was well conducted and well presented, although the findings support literature reports. The study limitations are also discussed. However, the role of comorbidities is not addressed in the manuscript. It is difficult to imagine that this cohort of patients have no other medical conditions that could contribute to or alter intensity of pain. Additionally, it would be interesting to note whether treatment success (e.g. lower IOP) correlates with presence of pain.

Author Response

We thank the reviewers for their appreciating comments and for their suggestion, which helped to improve our work. We did our best to address all points appropriately and we hope that you find our manuscript now suitable for publication.

However, the role of comorbidities is not addressed in the manuscript. It is difficult to imagine that this cohort of patients have no other medical conditions that could contribute to or alter intensity of pain.

Changes: (Discussion) We did not investigate other comorbidities causing pain. As the mean age was around 70 years there might have been several diseases, e.g. degenerative disorders of the spine, contributing to the intensity of pain. Since age was not different, this should have affected both groups in the same way. We did investigate preoperative pain and interestingly this was quite rare in both groups with 1.4% and 3% in the CW-group and in the MTSCPC-group respectively.

Additionally, it would be interesting to note whether treatment success (e.g. lower IOP) correlates with presence of pain.

Answer: For this study we did not evaluate follow-ups except for the first postoperative day. Here no difference in intraocular pressure was found between patients experiencing postoperative pain and those without pain.

Changes: (Results) No difference in intraocular pressure on the first postoperative day was found between patients experiencing pain with NRS ≥ 3 and other patients (23.8 [9.1] vs 22.3 [9.8] mmHg; p=0.310).